# GRAPHS, ENTITIES, AND STEP MIXTURE

## ABSTRACT

Graph neural networks have shown promising results on representing and analyzing diverse graph-structured data such as social, citation, and protein interaction networks. Existing approaches commonly suffer from the oversmoothing issue, regardless of whether policies are edge-based or node-based for neighborhood aggregation. Most methods also focus on transductive scenarios for fixed graphs, leading to poor generalization performance for unseen graphs. To address these issues, we propose a new graph neural network model that considers both edge-based neighborhood relationships and node-based entity features, i.e. **G**raph **E**ntities with **S**tep **M**ixture via *random walk* (GESM). GESM employs a mixture of various steps through random walk to alleviate the oversmoothing problem and attention to use node information explicitly. These two mechanisms allow for a weighted neighborhood aggregation which considers the properties of entities and relations. With intensive experiments, we show that the proposed GESM achieves state-of-the-art or comparable performances on four benchmark graph datasets comprising transductive and inductive learning tasks. Furthermore, we empirically demonstrate the significance of considering global information. The source code will be publicly available in the near future.

## 1 INTRODUCTION

Graphs are universal data representations that exist in a wide variety of real-world problems, such as analyzing social networks (Perozzi et al., 2014; Jia et al., 2017), forecasting traffic flow (Manley, 2015; Yu et al., 2017), and recommending products based on personal preferences (Page et al., 1999; Kim et al., 2019). Owing to breakthroughs in deep learning, recent graph neural networks (GNNs) (Scarselli et al., 2008) have achieved considerable success on diverse graph problems by collectively aggregating information from graph structures (Wang et al., 2018; Xu et al., 2018; Gao & Ji, 2019). As a result, much research in recent years has focused on how to aggregate the feature representations of neighbor nodes so that the dependence of graphs is effectively utilized.

The majority of studies have predominantly depended on edges to aggregate the neighboring nodes' features. These edge-based methods are premised on the concept of relational inductive bias within graphs (Battaglia et al., 2018), which implies that two connected nodes have similar properties and are more likely to share the same label (Kipf & Welling, 2017). While this approach leverages graphs' unique property of capturing relations, it appears less capable of generalizing to new or unseen graphs (Wu et al., 2019b).

To improve the neighborhood aggregation scheme, some studies have incorporated node information; They fully utilize node information and reduce the effects of relational (edge) information. A recent approach, graph attention networks (GAT), employs the attention mechanism so that weights used for neighborhood aggregation differ according to the feature of nodes (Veličković et al., 2018). This approach has yielded impressive performance and has shown promise in improving generalization for unseen graphs.

Regardless of neighborhood aggregation schemes, most methods, however, suffer from a common problem where neighborhood information is considered to a limited degree (Klicpera et al., 2019). For example, graph convolutional networks (GCNs) (Kipf & Welling, 2017) only operate on data that are closely connected due to oversmoothing (Li et al., 2018), which indicates the "washing out" of remote nodes' features via averaging. Consequently, information becomes localized and access to

global information is restricted (Xu et al., 2018), leading to poor performance on datasets in which only a small portion is labeled (Li et al., 2018).

In order to address the aforementioned issues, we propose a novel method, **G**raph **E**ntities with **S**tep **M**ixture via *random walk* (GESM), which considers information from all nodes in the graph and can be generalized to new graphs by incorporating *random walk* and *attention*. *Random walk* enables our model to be applicable to previously unseen graph structures, and a mixture of random walks alleviates the oversmoothing problem, allowing global information to be included during training. Hence, our method can be effective, particularly for nodes in the periphery or a sparsely labeled dataset. The *attention* mechanism also advances our model by considering node information for aggregation. This enhances the generalizability of models to diverse graph structures.

To validate our approach, we conducted experiments on four standard benchmark datasets: Cora, Citeseer, and Pubmed, which are citation networks for transductive learning, and protein-protein interaction (PPI) for inductive learning, in which test graphs remain unseen during training. In addition to these experiments, we verified whether our model uses information of remote nodes by reducing the percentage of labeled data. The experimental results demonstrate the superior performance of GESM on inductive learning as well as transductive learning for datasets. Moreover, our model achieved enhanced accuracy for datasets with reduced label rates, indicating the contribution of global information.

The key contributions of our approach are as follows:

- We present graphs with step mixture via *random walk*, which can adaptively consider local and global information, and demonstrate its effectiveness through experiments on public benchmark datasets with few labels.
- We propose **G**raph **E**ntities with **S**tep **M**ixture via *random walk* (GESM), an advanced model which incorporates attention, and experimentally show that it is applicable to both transductive and inductive learning tasks, for both nodes and edges are utilized for the neighborhood aggregation scheme.
- We empirically demonstrate the importance of propagation steps by analyzing its effect on performance in terms of inference time and accuracy.

## 2 BACKGROUNDS

### 2.1 RANDOM WALKS

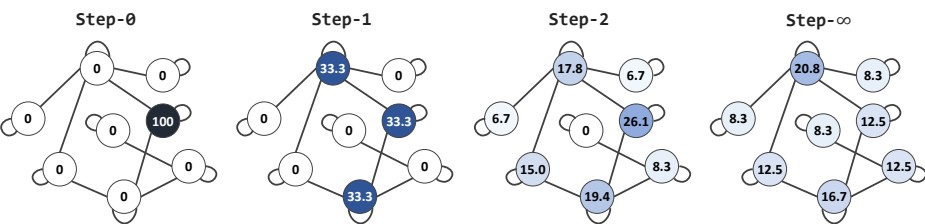

Figure 1: Random walk propagation procedure. From left to right are step-0, step-1, step-2, and step-infinite. The values in each node indicate the distribution of a random walk. In the leftmost picture, only the starting node has a value of 100, and all other nodes are initialized to zero. As the number of steps increases, values spread throughout the graph and converge to some extent.

Random walk, which is a widely used method in graph theory, mathematically models how node information propagates throughout the graph. As shown in Figure 1, random walk refers to randomly moving to neighbor nodes from the starting node in a graph. For a given graph, the transition matrix $P$, which describes the probabilities of transition, can be formulated as follows:

$$P = AD^{-1} \tag{1}$$

where $A$ denotes the adjacency matrix of the graph, and $D$ the diagonal matrix with a degree of nodes. The probability of moving from one node to any of its neighbors is equal, and the sum of the probabilities of moving to a neighboring node adds up to one.

Let $u^t$ be the distribution of the random walk at step $t$ ($u^0$ represents the starting distribution). The $t$ step random walk distribution is equal to multiplying $P$, the transition matrix, $t$ times. In other words,

$$u^1 = Pu^0$$
$$u^t = Pu^{t-1} = P^t u^0. \tag{2}$$

The entries of the transition matrix are all positive numbers, and each column sums up to one, indicating that $P$ is a matrix form of the Markov chain with steady-state. One of the eigenvalues is equal to 1, and its eigenvector is a steady-state (Strang, 1993). Therefore, even if the transition matrix is infinitely multiplied, convergence is guaranteed.

## 2.2 ATTENTION

The attention mechanism was introduced in sequence-to-sequence modeling to solve long-term dependency problems that occur in machine translation (Bahdanau et al., 2015). The key idea of attention is allowing the model to learn and focus on what is important by examining features of the hidden layer. In the case of GNNs (Scarselli et al., 2008), GATs (Veličković et al., 2018) achieved state-of-the-art performance by using the attention mechanism. Because the attention mechanism considers the importance of each neighboring node, node features are given more emphasis than structural information (edges) during the propagation process. Consequently, using attention is advantageous for training and testing graphs with different node features but the same structures (edges).

Given the many benefits of attention, we incorporate the attention mechanism to our model to fully utilize node information. The attention mechanism enables different importance values to be assigned to nodes of the same neighborhood, so combining attention with mixture-step random walk allows our model to adaptively highlight features with salient information in a global scope.

## 3 OUR PROPOSED METHODS

Let $\mathcal{G} = (V, E)$ be a graph, where $V$ and $E$ denote the sets of nodes and edges, respectively. Nodes are represented as a feature matrix $X \in \mathbb{R}^{n \times f}$, where $n$ and $f$ respectively denote the number of nodes and the input dimension per node. A label matrix is $Y \in \mathbb{R}^{n \times c}$ with the number of classes $c$, and a learnable weight matrix is denoted by $W$. The adjacency matrix of graph $\mathcal{G}$ is represented as $A \in \mathbb{R}^{n \times n}$. The addition of self-loops to the adjacency matrix is $\widetilde{A} = A + I_n$, and the column normalized matrix of $\widetilde{A}$ is $\hat{\widetilde{A}} = \widetilde{A}D^{-1}$ with $\hat{\widetilde{A}}^0 = I_n$.

### 3.1 GRAPH STEP MIXTURE: BASE APPROACH

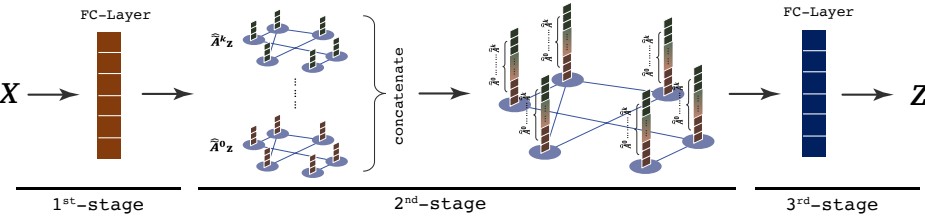

Figure 2: Schematic process of Graph Step Mixture. The procedure consists of three stages: (1st-stage) input passes the FC-layer, (2nd-stage) adjacency matrix is multiplied then concatenated, (3rd-stage) the final output is produced.

Most graph neural networks suffer from the oversmoothing issue along with localized aggregation. Although JK-Net (Xu et al., 2018) tried to handle oversmoothing by utilizing GCN blocks with mulitple propagation, it could not completely resolve the issue as shown in Figure 4b. We therefore propose **G**raph **S**tep **M**ixture (GSM), which not only separates the node embedding and propagation process but also tackles oversmoothing and localized aggregation issues through a mixture of random walk steps.

GSM has a simple structure that is composed of three stages, as shown in Figure 2. Input $X$ passes through a fully connected layer with a nonlinear activation. The output is then multiplied by a normalized adjacency matrix $\hat{\tilde{A}}$ for each random walk step that is to be considered. The results for each step are concatenated and fed into another fully connected layer, giving the final output. The entire propagation process of GSM can be formulated as:

$$Z_{\text{GSM}} = \text{softmax}\left(\left(\overset{s}{\underset{k=0}{\|}}\hat{\tilde{A}}^k \sigma(XW_0)\right)W_1\right),\tag{3}$$

where $\|$ is the concatenation operation, $s$ is the maximum number of steps considered for aggregation, and $\hat{\tilde{A}}^k$ is the normalized adjacency matrix $\hat{\tilde{A}}$ multiplied $k$ times. As can be seen from Equation 3, weights are shared across nodes.

In our method, the adjacency matrix $\hat{\tilde{A}}$ is an asymmetric matrix, which is generated by *random walks* and flexible to arbitrary graphs. On the other hand, prior methods such as JK-Net (Xu et al., 2018) and MixHop (Abu-El-Haija et al., 2019), use a symmetric Laplacian adjacency matrix, which limits graph structures to given fixed graphs.

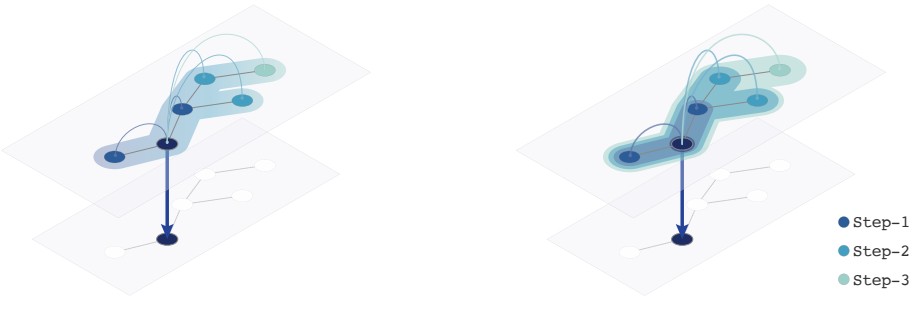

(a) Traditional global aggregation scheme  (b) Our step-mixture scheme

Figure 3: Conceptual scheme of neighborhood aggregation for three steps in conventional graph neural networks (a) and GSM which uses mixture of random walks (b).

For the concatenation operation, localized sub-graphs are concatenated with global graphs, which allows the neural network to adaptively select global and local information through learning (see Figure 3). While traditional graph convolution methods consider aggregated information within three steps by $A(A(AXW^{(0)})W^{(1)})W^{(2)}$, our method can take all previous aggregations into account by $(A^0XW \mid A^1XW \mid A^2XW \mid A^3XW)$.

### 3.2 GRAPH ENTITY STEP MIXTURE: GSM ENHANCED WITH ATTENTION

To develop our base model which depends on edge information, we additionally adopt the attention mechanism so that node information is emphasized for aggregation, i.e., **G**raph **E**ntity **S**tep **M**ixture (GESM). We simply modify the nonlinear transformation of the first fully connected layer in GSM by replacing it with the attention mechanism denoted by $H_{\text{multi}}$ (see Equations 3 and 4). As described in Equation 4, we employ multi-head attention, where $H_{\text{multi}}$ is the concatenation of $m$ attention layers and $\alpha$ is the coefficient of attention computed using concatenated features of nodes and its neighboring nodes.

$$Z_{\text{GESM}} = \text{softmax}\left(\left(\overset{s}{\underset{k=0}{\|}}\hat{\tilde{A}}^k H_{\text{multi}}\right)W_1\right), \qquad H_{\text{multi}} = \overset{m}{\underset{i=1}{\|}}\sigma(\alpha_i XW_0^i)\tag{4}$$

By incorporating attention to our base model, we can avoid or ignore noisy parts of the graph, providing a guide for random walk (Lee et al., 2018). Utilizing attention can also improve combinatorial generalization for inductive learning, where training and testing graphs are completely different. In particular, datasets with the same structure but different node information can benefit

Table 1: Summary of datasets used in the experiments.

| Type | Datasets | Nodes (Graphs) | | | | Features | Edges | Classes | Label rate |
|------|----------|-------|----------|------------|------|----------|-------|---------|------------|
| | | Total | Training | Validation | Test | | | | |
| T* | Cora | 2,708 | 140 | 500 | 1,000 | 1,433 | 5,429 | 7 | 5.1% |
| | Citeseer | 3,327 | 120 | 500 | 1,000 | 3,703 | 4,732 | 6 | 3.6% |
| | Pubmed | 19,717 | 60 | 500 | 1,000 | 500 | 44,338 | 3 | 0.3% |
| I† | PPI | 56,944 (24) | 44,906 (20) | 6,514 (2) | 5,524 (2) | 50 | 8,187 | 121 | - |

\* Transductive learning datasets consist of one graph and use a subset of the graph for training.
† Inductive learning datasets consist of many graphs and use a few graphs for training and unseen graphs for testing.

from our method because these datasets can only be distinguished by node information. Focusing on node features for aggregation can thus provide more reliable results in inductive learning.

The time complexity of our base model is $O(s \times l \times h)$, where $s$ is the maximum number of steps considered for aggregation, $l$ is the number of non-zero entries in the adjacency matrix, and $h$ is the hidden feature dimension. As suggested by Abu-El-Haija et al. (2019), we can assume $h << l$ under realistic assumptions. Our model complexity is, therefore, highly efficient with time complexity $O(s \times l)$, which is on par with vanilla GCN (Kipf & Welling, 2017).

## 4 EXPERIMENTS

### 4.1 DATASETS

***Transductive learning.*** We utilize three benchmark datasets for node classification: Cora, Citeseer, and Pubmed (Sen et al., 2008). These three datasets are citation networks, in which the nodes represent documents and the edges correspond to citation links. The edge configuration is undirected, and the feature of each node consists of word representations of a document. Detailed statistics of the datasets are described in Table 1.

For experiments on datasets with the public label rate, we follow the transductive experimental setup of Yang et al. (2016). Although all of the nodes' feature vectors are accessible, only 20 node labels per class are used for training. Accordingly, 5.1% for Cora, 3.6% for Citeseer, and 0.3% for Pubmed can be learned. In addition to experiments with public label rate settings, we conducted experiments using datasets where labels were randomly split into a smaller set for training. To check whether our model can propagate node information to the entire graph, we reduced the label rate of Cora to 3% and 1%, Citeseer to 1% and 0.5%, Pubmed to 0.1%, and followed the experimental settings of Li et al. (2018) for these datasets with low label rates. For all experiments, we report the results using 1,000 test nodes and use 500 validation nodes.

***Inductive learning.*** We use the protein-protein interaction PPI dataset (Zitnik & Leskovec, 2017) ,which is preprocessed by Veličković et al. (2018). As detailed in Table 1, the PPI dataset consists of 24 different graphs, where 20 graphs are used for training, 2 for validation, and 2 for testing. The test set remains completely unobserved during training. Each node is multi-labeled with 121 labels and 50 features regarding gene sets and immunological signatures.

### 4.2 COMPARISON MODELS

For transductive learning, we compare our model with numbers of state-of-the-art models according to the results reported in the corresponding papers. Our model is compared with baseline models specified in (Veličković et al., 2018) such as label propagation (LP) (Xiaojin & Zoubin, 2002), graph embeddings via random walk (DeepWalk) (Perozzi et al., 2014), and Planetoid (Yang et al., 2016). We also compare our model with models that use self-supervised learning (Union) (Li et al., 2018), learnable graph convolution (LGCN) (Gao et al., 2018), GCN based multi-hop neighborhood mixing (JK-GCN and MixHop) (Xu et al., 2018; Abu-El-Haija et al., 2019), multi-scale graph convolutional networks (AdaLNet) (Liao et al., 2019) and maximal entropy transition (PAN) (Ma et al., 2019). We

Table 2: Experimental results on the public benchmark datasets. Evaluation metrics on transductive and inductive learning datasets are classification accuracy (%) and F1-score, respectively. Top-3 results for each column are highlighted in bold, and top-1 values are underlined.

| | Transductive | | | Inductive |
| | Cora public (5.1%) | Citeseer public (3.6%) | Pubmed public (0.3%) | PPI |
| Method | | | | |
|---|---|---|---|---|
| LP (Xiaojin & Zoubin, 2002) | 68.0 | 45.3 | 63.0 | - |
| Deep Walk (Perozzi et al., 2014) | 67.2 | 43.2 | 65.3 | - |
| Cheby (Defferrard et al., 2016) | 81.2 | 69.8 | 74.4 | - |
| Planetoid (Yang et al., 2016) | 75.7 | 64.7 | 77.2 | - |
| GCN (Kipf & Welling, 2017) | 81.5 | 70.3 | 79.0 | - |
| GraphSAGE (Hamilton et al., 2017) | - | - | - | 0.612 |
| GAT (Veličković et al., 2018) | 83.0 | **72.5** | 79.0 | **0.973** |
| LGCN (Gao et al., 2018) | **83.3** | **73.0** | 79.5 | 0.772 |
| JK-LSTM (Xu et al., 2018) | - | - | - | **0.976** |
| AGNN (Thekumparampil et al., 2018) | 83.1 | 71.7 | 79.9 | - |
| Union (Li et al., 2018) | 80.5 | 65.7 | 78.3 | - |
| *APPNP (Klicpera et al., 2019) | **83.2** | 71.7 | 79.7 | - |
| SGC (Wu et al., 2019a) | 81.0 | 71.9 | 78.9 | - |
| MixHop (Abu-El-Haija et al., 2019) | 81.9 | 71.4 | **80.8** | - |
| GWNN (Xu et al., 2019) | 82.8 | 71.7 | 79.1 | - |
| AdaLNet (Liao et al., 2019) | 80.4 | 68.7 | 78.1 | - |
| PAN (Ma et al., 2019) | 82.0 | 71.2 | 79.2 | - |
| GSM (our base model) | 82.8 | 71.7 | **80.3** | 0.753 |
| GESM (GSM+attention) | **84.4** | 72.6 | 80.1 | **0.976** |

* Best experimental results through our own implementation

further include models that utilize teleport term during propagation APPNP (Klicpera et al., 2019), conduct convolution via spectral filters such as ChebyNet (Defferrard et al., 2016), GCN (Kipf & Welling, 2017), SGC (Wu et al., 2019a), and GWNN (Xu et al., 2019) and models that adopt attention between nodes, such as GAT (Veličković et al., 2018) and AGNN (Thekumparampil et al., 2018).

For inductive learning tasks, we compare our model against four baseline models. This includes graphs that use sampling and aggregation (GraphSAGE-LSTM) (Hamilton et al., 2017), and jumping-knowledge (JK-LSTM) (Xu et al., 2018), along with GAT and LGCN which are used in the transductive setting.

## 4.3 EXPERIMENTAL SETUP

Regarding the hyperparameters of our transductive learning models, we used different settings for datasets with the public split and random split. We set the dropout probability such that 0.3 of the data were kept for the public split and 0.6 were kept for the random split. We set the number of multi-head $m = 8$ for GESM. The size of the hidden layer $h \in \{64, 512\}$ and the maximum number of steps used for aggregation $s \in \{10, 30\}$ were adjusted for each dataset. We trained for a maximum of 300 epochs with L2 regularization $\lambda = 0.003$ and learning rate $lr = 0.001$. We report the average classification accuracy of 20 runs.

For inductive learning, the size of all hidden layers was the same with $h = 256$ for both GSM, which consisted of two fully connected layers at the beginning and GESM. We set the number of steps $s = 10$ for GSM, and $s = 5, m = 15$ for GESM. L2 regularization and dropout were not used for inductive learning (Veličković et al., 2018). We trained our models for a maximum of 2,000 epochs with learning rate $lr = 0.008$. The evaluation metric was the micro-F1 score, and we report the averaged results of 10 runs.

For all the models, the nonlinearity function of the first fully connected layer was an exponential linear unit (ELU) (Clevert et al., 2016). Our models were initialized using Glorot initialization (Glorot & Bengio, 2010) and were trained to minimize the cross-entropy loss using the Adam

Table 3: Node classification results on datasets with low label rates. Top-3 results for each column are highlighted in bold and top-1 values are underlined.

| Method | Cora | | Citeseer | | Pubmed |
|---|---|---|---|---|---|
| | 1% | 3% | 0.5% | 1% | 0.1% |
| LP (Xiaojin & Zoubin, 2002) | 62.3 | 67.5 | 34.8 | 40.2 | 65.4 |
| Cheby (Defferrard et al., 2016) | 52.0 | 70.8 | 31.7 | 42.8 | 51.2 |
| GCN (Kipf & Welling, 2017) | 62.3 | 76.5 | 43.6 | 55.3 | 65.9 |
| Union (Li et al., 2018) | **69.9** | 78.5 | **46.3** | 59.1 | 70.7 |
| *JK-GCN (Xu et al., 2018) | 65.1 | 76.8 | 37.1 | 55.3 | 71.1 |
| *APPNP (Klicpera et al., 2019) | 67.6 | **80.8** | 40.5 | 59.9 | 70.7 |
| *SGC (Wu et al., 2019a) | 64.2 | 77.2 | 41.0 | 58.1 | 71.7 |
| AdaLNet (Liao et al., 2019) | 67.5 | 77.7 | **53.8** | **63.3** | **72.8** |
| GSM (our base model) | **68.2** | **81.6** | 45.6 | **62.6** | **73.0** |
| GESM (GSM+attention) | **70.5** | **81.2** | **53.2** | **62.7** | **73.8** |

* Best experimental results through our own implementation

optimizer (Kingma & Ba, 2015). We employed an early stopping strategy based on the loss and accuracy of the validation sets, with a patience of 100 epochs.

## 5 RESULTS

### 5.1 NODE CLASSIFICATION

***Results on benchmark datasets.*** Table 2 summarizes the comparative evaluation experiments for transductive and inductive learning tasks. In general, not only are there a small number of methods that can perform on both transductive and inductive learning tasks, but the performance of such methods is not consistently high. Our methods, however, are ranked in the top-3 for every task, indicating that our method can be applied to any task with large predictive power.

For transductive learning tasks, the experimental results of our methods are higher than or equivalent to those of other methods. As can be identified from the table, our base model GSM, which is computationally efficient and simple, outperforms many existing baseline models. These results indicate the significance of considering both global and local information and using random walks. It can also be observed that GESM yielded more impressive results than GSM, suggesting the importance of considering node information in the aggregation process.

For the inductive learning task, our base model GSM, which employs an edge-based aggregation method, does not invariably obtain the highest accuracy. However, our model with attention, GESM, significantly improves performance of GSM by learning the importance of neighborhood nodes, and surpasses the results of GAT, despite the fact that GAT consists of more attention layers. These results for unseen graphs are in good agreement with results shown by Veličković et al. (2018), in which reducing the influence of structural information improved generalization.

***Results on datasets with low label rates.*** To demonstrate that our methods can consider global information, we experimented on sparse datasets with low label rates of transductive learning datasets. As indicated in Table 3, our models show remarkable performance even on the dataset with low label rates. In particular, we can further observe the superiority of our methods by inspecting Table 2 and 3, in which our methods trained on only 3% of the Cora dataset outperformed some other methods trained on 5.1% of the data. Because both GSM and GESM showed enhanced accuracy, it could be speculated that using a mixture of random walks played a key role in the experiments; the improved results can be explained by our methods adaptively selecting node information from local and global neighborhoods, and allowing peripheral nodes to receive information.

### 5.2 MODEL ANALYSIS

***Oversmoothing and Accuracy.*** As shown in Figure 4a, GCN (Kipf & Welling, 2017), SGC (Wu et al., 2019a), and GAT (Veličković et al., 2018) suffer from oversmoothing. GCN and GAT show

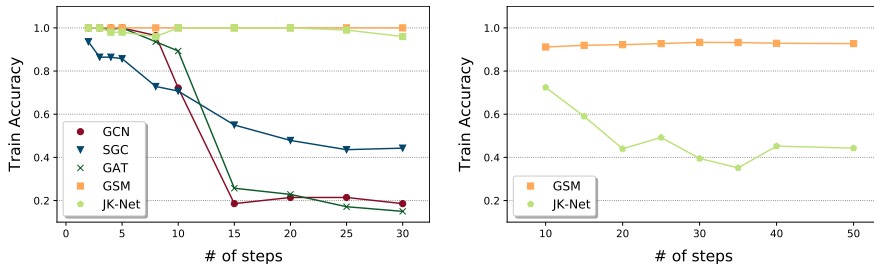

(a) Training accuracy comparison according to number of propagation steps.

(b) Test accuracy of JK-Net and GSM using concatenated features after the 10th step.

Figure 4: Effect of the step size on model performance on Cora.

severe degradation in accuracy after the 8th step; The accuracy of SGC does not drop as much as GCN and GAT but nevertheless gradually decreases as the step size increases. The proposed GSM, unlike the others, maintains its performance without any degradation, because no rank loss (Luan et al., 2019) occurs and oversmoothing is overcome by step mixture.

Interestingly, JK-Net (Xu et al., 2018) also keeps the training accuracy regardless of the step size by using GGN blocks with multiple steps according to Figure 4a. We further compared the test accuracy of GSM with JK-Net, a similar approach to our model, in regards to the step size. To investigate the adaptability to larger steps of GSM and JK-Net, we concatenated features after the 10th step. As shown in Figure 4b, GSM outperforms JK-Net, even though both methods use concatenation to alleviate the oversmoothing issue. These results are in line with the fact that JK-Net obtains global information similar to GCN or GAT. Consequently, the larger the step, the more difficult it is for JK-Net to maintain performance. GSM, on the other hand, maintains a steady performance, which confirms that GSM does not collapse even for large step sizes.

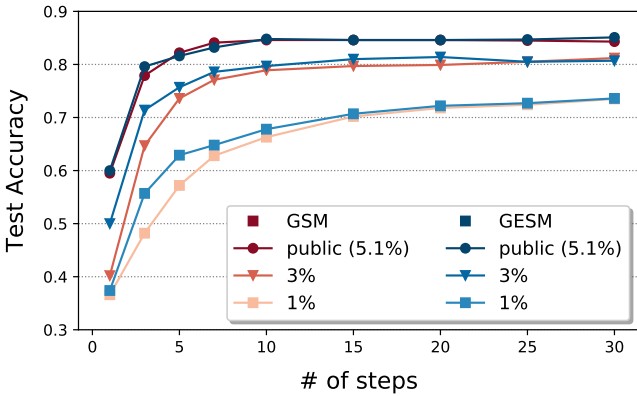

Figure 5: Accuracy comparison for various label rates according to step size.

We also observe the effect on accuracy as the number of steps increases under three labeling conditions for GSM and GESM. As represented in Figure 5, it is evident that considering remote nodes can contribute to the increase in accuracy. By taking into account more data within a larger neighborhood, our model can make reliable decisions, resulting in improved performance. Inspection of the figure also indicates that the accuracy converges faster for datasets with higher label rates, presumably because a small number of walk steps can be used to explore the entire graph. Moreover, the addition of attention benefits performance in terms of higher accuracy and faster convergence.

***Inference time.*** As shown in Figure 6, the computational complexity of all models increases linearly as the step size increases. We can observe that the inference time of GSM is faster than GCN (Kipf & Welling, 2017) especially when the number of steps is large. The inference time of GESM is much faster than GAT (Veličković et al., 2018) while providing higher accuracies and stable results (see

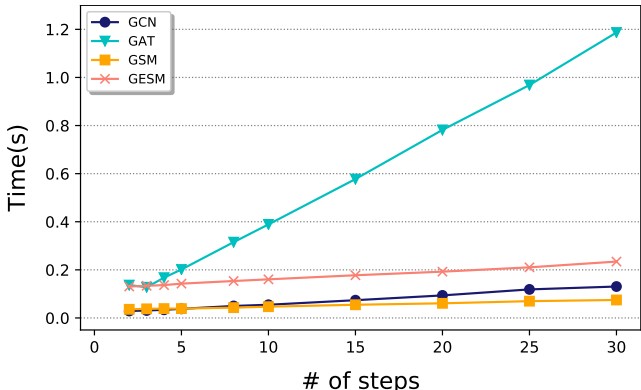

Figure 6: Inference time of various models as the step size increases for Cora dataset.

Appendix A). Our methods are both fast and accurate due to the sophisticated design with a mixture of random walk steps.

***Embedding Visualization.*** Figure 7 visualizes the hidden features of Cora from our models by using

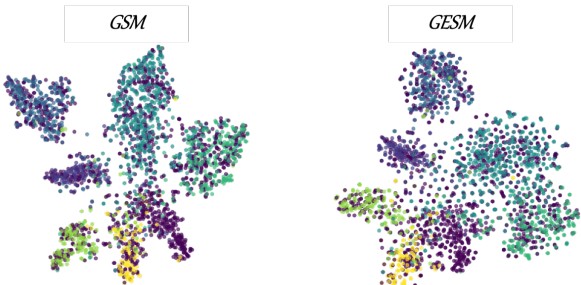

Figure 7: t-SNE plot of the last hidden layer trained on the Cora dataset.

the t-SNE algorithm (Maaten & Hinton, 2008). The figure illustrates the difference between edge-based and node-based aggregation. While the nodes are closely clustered in the result from GSM, they are scattered in that of GESM. According to the results in Table 2, more closely clustered GSM does not generally produce better results than loosely clustered GESM, which supports findings that the attention mechanism aids models to ignore or avoid noisy information in graphs (Lee et al., 2018).

## 6 CONCLUSION

In this paper, we have proposed simple but effective two types of models to utilize global information and improve generalization for unseen graphs. GSM, the base model, is computationally efficient and effectively aggregate global and local information by employing multiple random walks. To further refine our base model, we have presented GESM, which explicitly leverages node information (see Appendix B). The results from extensive experiments show that our models successfully achieve state-of-the-art or competitive performance for both transductive and inductive learning tasks on four benchmark graph datasets. As future directions, we will refine our method that utilizes node information to improve the computational efficiency regarding attention. In addition, we will extend GESM so that it can be applied to real-world large scale graph data.

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

Table 4: Summary of datasets used in the additional experiments.

|  | Classes | Features | Nodes | Edges | Label rate |
|---|---|---|---|---|---|
| Coauthor CS | 15 | 6,805 | 18,333 | 100,227 | 1.6% |
| Coauthor Physics | 5 | 8,415 | 34,493 | 282,455 | 0.2% |
| Amazon Computers | 10 | 767 | 13,381 | 259,159 | 1.4% |
| Amazon Photo | 8 | 745 | 7,487 | 126,530 | 2.1% |

Table 5: Average test set accuracy and standard deviation over 100 random train/validation/test splits with 20 runs. Top-3 results for each column are highlighted in bold, and top-1 values are underlined.

|  | Coauthor CS | Coauthor Physics | Amazon Computers | Amazon Photo |
|---|---|---|---|---|
| MLP | $88.3 \pm 0.7$ | $88.9 \pm 1.1$ | $44.9 \pm 5.8$ | $69.6 \pm 3.8$ |
| LogReg | $86.4 \pm 0.9$ | $86.7 \pm 1.5$ | $64.1 \pm 5.7$ | $73.0 \pm 6.5$ |
| LP (Xiaojin & Zoubin, 2002) | $73.6 \pm 3.9$ | $86.6 \pm 2.0$ | $70.8 \pm 8.1$ | $72.6 \pm 11.1$ |
| GCN (Kipf & Welling, 2017) | $91.1 \pm 0.5$ | $92.8 \pm 1.0$ | $\mathbf{82.6} \pm 2.4$ | $\mathbf{91.2} \pm 1.2$ |
| GraphSAGE (Hamilton et al., 2017) | $\mathbf{91.3} \pm 2.8$ | $\mathbf{93.0} \pm 0.8$ | $\mathbf{82.4} \pm 1.8$ | $\underline{\mathbf{91.4}} \pm 1.3$ |
| GAT (Veličković et al., 2018) | $90.5 \pm 0.6$ | $92.5 \pm 0.9$ | $78.0 \pm 19.0$ | $85.7 \pm 20.3$ |
| GSM (our base model) | $\mathbf{91.8} \pm 0.4$ | $\mathbf{93.3} \pm 0.6$ | $79.2 \pm 2.0$ | $89.3 \pm 1.9$ |
| GESM (GSM+attention) | $\underline{\mathbf{92.0}} \pm 0.5$ | $\underline{\mathbf{93.7}} \pm 0.6$ | $\mathbf{79.3} \pm 1.7$ | $\mathbf{90.0} \pm 2.0$ |

## A EXPERIMENTS ON OTHER DATASETS

For an in-depth verification of overfitting, we extended our experiments to four types of new node classification datasets. Coauthor CS and Coauthor Physics are co-authorship graphs from the KDD Cup 2016 challenge[1], in which nodes are authors, features represent the article keyword for each author's paper, and class labels indicate each author's most active research areas. Amazon Computers and Amazon Photo are co-purchase graphs of Amazon, where nodes represent the items, and edges indicate that items have been purchased together. The node features are bag-of-words of product reviews, and class labels represent product categories. Detailed statistics of the datasets are described in Table 4 and we followed the experimental setup of Shchur et al. (2018).

We used the same values for each hyperparameter (unified size: 64, step size: 15, multi-head for GAT and GESM: 8) without tuning. The results in Table 5 prove that our proposed methods do not overfit to a particular dataset. Moreover, in comparison to GAT, the performance of GESM is more accurate, and more stable.

## B ATTENTION VISUALIZATION

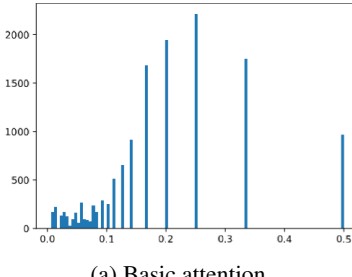

(a) Basic attention.

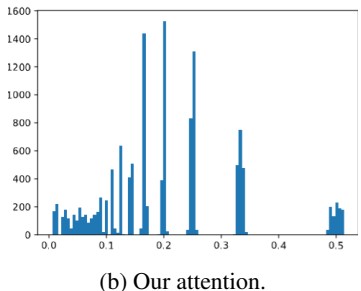

(b) Our attention.

Figure 8: Visualization of attention distribution.

---

[1]https://kddcup2016.azurewebsites.net/

We visualized the distribution of attention vectors. Figure 8a plots the distribution of neighbors with equal importance and Figure 8b displays the distribution of attention weighted neighbors that we trained with GESM. Although both figures look similar to some degree, we can conjecture that GESM slightly adjusts the weight values, contributing to improved performance.

