# OpenReview forum: "GRAPHS, ENTITIES, AND STEP MIXTURE"
_ICLR.cc/2020/Conference — Reject_

### Official Review · AnonReviewer1 · 2019-10-21
**Official Blind Review #1**

**Rating:** 6

**Review:**

The paper presents a graph neural network model that aims at improving the feature aggregation scheme to better handle distant nodes, therefore mitigating the “smoothing” problem of classic averaging.

I find the paper clearly motivated and easy to follow, although some sentences could be streamlined and some repetitions could be removed. For instance last paragraph in page 3, this should be clear already and should be restated.

One thing that is not clear is how does the model cope with the increase in feature dimensionality due to the concatenation over different steps. Isn’t this leading to overfitting? Did the authors experiment with other schemes such as averaging or gating? If so it would be nice to see the results for each of the configurations as it is not clear to me what should be chosen a-priori.

Experiments are nicely executed and the proposed approach is compared against a rich array of other models. Results are state-of-the-art and also the analysis of the model is interesting, i.e. it doesn’t diverge when increasing # steps at test time.
How does the attention vector look like? Does it tend to peak at a given k, or is it more uniformly distributed?

How does the model compare to having k GAT layers, each constrained to use neighboring nodes at step k as input for the attention computation? Did the authors experiment on this?

Overall I like the work but find the novelty quite limited, more effort could have been put into motivating the soundness of the use of multiple random walks. Perhaps some theory could be developed to make the paper stronger.

**Experience Assessment:**

I have published in this field for several years.

**Review Assessment: Checking Correctness Of Derivations And Theory:**

I carefully checked the derivations and theory.

**Review Assessment: Checking Correctness Of Experiments:**

I carefully checked the experiments.

**Review Assessment: Thoroughness In Paper Reading:**

I read the paper at least twice and used my best judgement in assessing the paper.

---

> ### Author Response · Authors · 2019-11-11
> **Author response to Review #1**
>
> We appreciate R1 for the clear but rich insightful suggestion. Thanks to R1's constructive feedback, we can find the few things in our study that need to be discussed.
>
> The responses for each comment are as follows:
>
>
> 1. some sentences could be streamlined and some repetitions could be removed. For instance last paragraph in page 3, this should be clear already and should be restated.
>
> A)
> Thanks for the advice. We will revise it later and reflect it in the paper.
>
>
> 2. One thing that is not clear is how does the model cope with the increase in feature dimensionality due to the concatenation over different steps. Isn’t this leading to overfitting? Did the authors experiment with other schemes such as averaging or gating? If so it would be nice to see the results for each of the configurations as it is not clear to me what should be chosen a-priori.
>
> A)
> As represented in Figure 4b (Figure 5 in revision), the test predictions converge to a certain level as the number of steps increases. Although the parameters increase as the steps increase, there is no decrease in performance due to overfitting.
>
> We tried averaging scheme as R1 mentioned but the results were not as good as concatenation scheme (average pooling: 72.5%, max pooling: 77.3%).
>
>
> 3. Experiments are nicely executed and the proposed approach is compared against a rich array of other models. Results are state-of-the-art and also the analysis of the model is interesting, i.e. it doesn’t diverge when increasing # steps at test time.
>
> A)
> Thanks for your kind comments. The number of parameters and overfitting in graph neural networks are very important issues. We will study it in future research.
>
>
> 4. How does the attention vector look like? Does it tend to peak at a given k, or is it more uniformly distributed?
>
> A)
> Thanks for the insightful suggestion. There was no big difference in attention distribution depending on the steps. However, we found that GESM slightly adjusts the weight values compared to the uniform distribution and contributed to performance improvement. We will attach these results to the Appendix.
>
>
> 5. How does the model compare to having k GAT layers, each constrained to use neighboring nodes at step k as input for the attention computation? Did the authors experiment on this?
>
> A)
> (Test ACC)
> +---------------------------------+------------------+-------------------------+---------------------+--------------+
> | Model/Test ACC             | Coauthor CS | Coauthor Physics | Amazon Computers | Amazon Photo |
> +---------------------------------+------------------+-------------------------+---------------------+--------------+
> | GCN[1]                             |    91.1±0.5   |          92.8±1.0         |        82.6±2.4      |     91.2±1.2   |
> | GAT[2]                              |    90.5±0.6   |          92.5±0.9         |        78.0±19.0    |     85.7±20.3 |
> | GSM (our base)              |     91.8±0.4   |          93.3±0.6        |        79.2±2.1      |     89.3±1.9    |
> | GESM (GSM+attention) |     92.0±0.5   |          93.7±0.6        |        79.3±1.7      |     90.0±2.0   |
> +---------------------------------+-----------------+--------------------------+---------------------+---------------+
>
>  (Inference Time(s))
> +-----------------------+---------+----------+---------+---------+---------+
> |    Model/Step     |     2     |     5     |     8      |   15    |   20    |
> +-----------------------+---------+----------+---------+---------+---------+
> | GCN[1]                | 0.028  | 0.037  | 0.049  | 0.073 | 0.092 |
> | GAT[2]                 | 0.136  | 0.201  | 0.315  | 0.577 | 0.781 |
> | GSM                    | 0.035  | 0.039  | 0.043  | 0.060 | 0.071 |
> | GESM                  | 0.131  | 0.143  | 0.153  | 0.178 | 0.211 |
> +-----------------------+---------+---------+----------+---------+---------+
>
> Thanks for the good suggestion. Both of the above experiments were performed with the same hyper parameter under the same conditions with layer: 64 multi head: 8. We can see that our models are faster and stable (A new experiment is conducted on [3] by the suggestion of Reviewer 2).
>
>
> 6. Overall I like the work but find the novelty quite limited, more effort could have been put into motivating the soundness of the use of multiple random walks. Perhaps some theory could be developed to make the paper stronger.
>
> A)
> A low-pass filter of GCN [1] matrix has a problem in generalization because it is based on a laplacian eigen basis in spectral domain [4]. Random walk, on the other hand, is not a methodology in spectral domain, so it can be easily applied to multiple graphs.
>
>
> We will update our manuscript by reflecting the comments and responses as soon as possible.
>
> [1] Kipf and Welling: Semi-Supervised Classification with Graph Convolutional Networks
> [2] Petar Veličković et al: Graph Attention Networks
> [3] Shchur et al.: Pitfalls of Graph Neural Network Evaluation
> [4] Zonghan Wu et al. : A Comprehensive Survey on Graph Neural Networks

---

### Official Review · AnonReviewer2 · 2019-10-23
**Official Blind Review #2**

**Rating:** 3

**Review:**

This paper presents two models, namely GSM and GESM, to tackle the problem of transductive and inductive node classification. GSM is operating on asymmetric transition matrices and works by stacking propagation layers of different locality, where the final prediction is based on all propagation steps (JK concatenation style). GESM builds upon GSM and introduces a multi-headed attention layer applied on the initial feature matrix to guide the propagation layers. The models are evaluated on four common benchmark datasets and achieve state-of-the-art performance, especially when the training label rate is reduced.

Overall, the paper is well-written and its presentation is mostly clear and comprehensible (see below). The quantitative evaluation looks good to me, especially since an ablation study shows the contributions of all of the proposed features.

However, there are a few weak points which should explain my overall score:

1. The proposed GSM model is not new and only re-uses building blocks from the related work. [1] shows that removing non-linearities is an effective procedure for node classification. [2] investigates the massively stacking of propagations. The procedure of feature concatenation from different locality has been studied in [3]. Applying asymmetric normalization is a standard aggregation scheme for GNNs, e.g., in [4].

2. The GESM model is not fully understandable since it is missing a formal description for computing $\alpha$. It is only said that $\alpha$ is computed using the concatenation of features from the central node and its neighbors. Can you elaborate how exactly you compute $\alpha$, especially since the concatenation of neighboring features results in a non-permutation invariant architecture? In addition, in contrast to the reported results in Tables 3 and 4, Figure 4 indicates that the benefits of GESM are negligible.

3. The final prediction layer with weight matrix W_1 operates on all propagation layers, resulting in a parameter complexity of $O(s  h c)$, where $c$ is the number of classes. With $s=30$ and $h=512$, this results in 15.360  c parameters (!!!), whereas GCN [5] only uses 16  c parameters. Hence, I do not think it is fair to promote your model as efficient as vanilla GCN. In addition, the final matrix multiplication results in a computational complexity of $O(n  s^2  h^2  c)$ which does not nearly match your reported complexity.
Furthermore, I do wonder why your model is not heavily overfitting with such an amount of parameters. For example, this is the reason [3] does evaluate its model on a larger training split instead of a smaller one.

4. As your work is quite similar to [1, 2], it would be beneficial to also include the respective results of those methods in Tables 2 and 3. In addition, their differences and similarities should be discussed in detail.

5. Since the used benchmark datasets are already reasonably explored, authors are advised to include evaluation on other datasets as well, e.g., from [6].

6. The transition matrix $P$ is missing self-loops to match with the results of Figure 1. Since you already define $P$ analogously to $\hat{\tilde{A}}$, you should focus on one notation for consistency reasons.

[1] Wu et al.: Simplifying Graph Convolutional Networks
[2] Klicpera et al.: Predict then Propagate: Graph Neural Networks meet Personalized PageRank
[3] Xu et al.: Representation Learning on Graphs with Jumping Knowledge Networks
[4] Hamilton et al.: Inductive Representation Learning on Large Graphs
[5] Kipf and Welling: Semi-Supervised Classification with Graph Convolutional Networks
[6] Shchur et al.: Pitfalls of Graph Neural Network Evaluation

----------------------------
Update after the rebuttal: The authors have addressed several issues and improved their manuscript. I greatly appreciate the effort and the new experimental results. However, the main weak point that the novelty of the approach is limited remains valid of course. Therefore, I am still more inclined to rejecting the paper. I have raised my score from "1: Reject" to "3: Weak Reject".

I have raised my score from "5: Weak Reject" to "6: Weak Accept".

**Experience Assessment:**

I have published one or two papers in this area.

**Review Assessment: Checking Correctness Of Derivations And Theory:**

I carefully checked the derivations and theory.

**Review Assessment: Checking Correctness Of Experiments:**

I assessed the sensibility of the experiments.

**Review Assessment: Thoroughness In Paper Reading:**

I read the paper thoroughly.

---

> ### Author Response · Authors · 2019-11-11
> **Author response to Review #2 (part1)**
>
> We appreciate R2 for the rich advice. Thanks to your constructive feedback, we were able to rethink about the shortcomings, and it has helped us improve the quality of our research.
>
> The responses for each comment are as follows:
>
>
> 1. The proposed GSM model is not new and only re-uses building blocks from the related work. [1] shows that removing non-linearities is an effective procedure for node classification. [2] investigates the massively stacking of propagations. The procedure of feature concatenation from different locality has been studied in [3]. Applying asymmetric normalization is a standard aggregation scheme for GNNs, e.g., in [4].
>
> A)
> We agree that our method, an attention-enhanced mixture of random work, is based on a simple graph block and the novelty might be not very large. However, the relevant methods, including ours, showed noticeably different results for the oversmoothing issue, which has not been resolved yet. Unlike other methods, our method is consistently superior to others in prediction accuracy and computation time.
>
> [Table 1-1] To check the oversmoothing issue (Train ACC)
> +-----------------+-------+-------+-------+-------+-------+
> | model/step |   2   |    5   |    8   |   15   |   20  |
> +-----------------+-------+-------+-------+-------+-------+
> | GCN[4]        | 1.00 | 1.00 | 0.92 | 0.23 | 0.20 |
> | GAT[5]         | 1.00 | 1.00 | 0.89 | 0.25 | 0.22 |
> | SGC[1]         | 1.00 | 1.00 | 0.87 | 0.77 | 0.72 |
> | GSM (ours) | 1.00 | 1.00 | 1.00 | 1.00 | 1.00 |
> +-----------------+-------+-------+-------+-------+-------+
>
> As shown in Table 1-1, SGC[1], GCN[4], and GAT[5] suffer from oversmoothing issue. GCN and GAT show severe degradation in accuracy since the 8th step; Even if SGC is better than GCN and GAT, its accuracy continues to decrease as the step size increases. Therefore JK-Net[3], which is based on GCN or GAT propagation, does not seem to utilize global information properly. Our proposed method GSM maintains its accuracy even in global steps without degradation.
>
> [Table 1-2] To check the global aggregation (Average Test ACC of 10 runs)
> +-----------------------+--------+--------+--------+--------+--------+
> | Model/Step        |   10   |   15   |    20   |    25  |   30   |
> +-----------------------+--------+--------+--------+--------+--------+
> | JK-Net[3]             |  0.44 |  0.40 |  0.30 |  0.27 |  0.24 |
> | GSM (our base) |  0.80 |  0.81 |  0.81 |  0.81 |  0.81 |
> +---------------------- +--------+--------+--------+--------+--------+
>
> For more detailed comparison ours with JK-Net about global information, we checked test accuracy by storing features after the 10th step. Both methods are based on concatenation to alleviate the oversmoothing issue, but as the underlying model differs, eventually the test accuracy is significantly different as represented in Table 1-2. Since global information of JK-Net is obtained from GCN or GAT, the longer the step, the more difficult it is to maintain performance. GSM, on the other hand, keeps steady performance, which proves that GSM does not collapse even in global steps.
>
> In addition, APPNP [2] is Neumann series approximation algorithms [7], which can be considered as a simple sum of steps, and it is inferior in performance to ours as shown in various experimental results.
>
> Therefore, this is not a trivial approach because other models that share some relevant ideas do not show competitive performance compared to ours.
>
>
> 2. The GESM model is not fully understandable since it is missing a formal description for computing $\alpha$. It is only said that $\alpha$ is computed using the concatenation of features from the central node and its neighbors. Can you elaborate how exactly you compute $\alpha$, especially since the concatenation of neighboring features results in a non-permutation invariant architecture? In addition, in contrast to the reported results in Tables 3 and 4, Figure 4 indicates that the benefits of GESM are negligible.
>
> A)
> Sorry for missing a formal description the attention
> $\alpha = \text{softmax}(W1H1+W2H2)$,  where $H1$ denotes a central node, $H2$ denotes a neighbor node.
>
> In order to maintain the permutation invariant, $\alpha$ is computed based on only one direction of nodes (keeping the order of nodes).
>
> The results in Figure 4b (Figure 5 in revision) might be slightly different from the results in the table because we do not use early stops and averaging of multiple runs used on the table. But GESM converges faster than GSM and has overwhelming results in inductive learning.

---

> ### Author Response · Authors · 2019-11-11
> **Author response to Review #2 (part2)**
>
> 3. The final prediction layer with weight matrix W_1 operates on all propagation layers, resulting in a parameter complexity of , where is the number of classes. With and , this results in 15.360  c parameters (!!!), whereas GCN [5] only uses 16  c parameters. Hence, I do not think it is fair to promote your model as efficient as vanilla GCN. In addition, the final matrix multiplication results in a computational complexity of which does not nearly match your reported complexity. Furthermore, I do wonder why your model is not heavily overfitting with such an amount of parameters. For example, this is the reason [3] does evaluate its model on a larger training split instead of a smaller one.
>
> A)
> [Table 3-1] Inference Time(s)
> +-----------------------+---------+----------+---------+---------+---------+
> |    Model/Step     |     2     |     5     |     8      |   15    |   20    |
> +-----------------------+---------+----------+---------+---------+---------+
> | GCN[4]                | 0.028  | 0.037  | 0.049  | 0.073 | 0.092 |
> | GAT[5]                 | 0.136  | 0.201  | 0.315  | 0.577 | 0.781 |
> | GSM                    | 0.035  | 0.039  | 0.043  | 0.060 | 0.071 |
> | GESM                  | 0.131  | 0.143  | 0.153  | 0.178 | 0.211 |
> +-----------------------+---------+---------+----------+---------+---------+
>
> [Table 3-2] Test ACC on new datasets
> +---------------------------------+------------------+--------------------------+-----------------------------+-----------------------+
> | Model/Test ACC             | Coauthor CS | Coauthor Physics | Amazon Computers | Amazon Photo |
> +---------------------------------+------------------+--------------------------+-----------------------------+-----------------------+
> | GCN[4]                             |     91.1±0.5   |           92.8±1.0         |            82.6±2.4           |        91.2±1.2       |
> | GAT[5]                              |     90.5±0.6   |           92.5±0.9         |            78.0±19.0         |        85.7±20.3    |
> | GSM (our base)              |     91.8±0.4   |           93.3±0.6         |            79.2±2.1           |        89.3±1.9       |
> | GESM (GSM+attention) |     92.0±0.5   |           93.7±0.6         |            79.3±1.7           |        90.0±2.0      |
> +---------------------------------+-----------------+---------------------------+-----------------------------+----------------------+
>
> We agree that our method uses more parameter for feature concatenation, thus leading to more computation in the last layer. However, this computation cost does not severely harm the real inference time.
>
> To check a computational complexity, we measured an inference time on Cora dataset. Due to the realistic assumption written in the manuscript (hidden size << non zero entities), the experimental computation complexity increases linearly[8] with respect to steps as shown in Table 3-1 and in the revised manuscript Figure 6. The inference time of GSM is less than GCN in longer steps, and the time of GESM is much faster than GAT while providing higher accuracies.
>
> To check about overfitting, we carried out additional experiments on extensive number of training splits as you mentioned. We used unified number of parameters (unified size: 64, step size: 15)  without any special parameter tuning. As shown in Table 3-2, the proposed approaches showed robust performance even in the new datasets.
>
> For a fair comparison, we conducted experiments by reducing the number of parameters used in our method similar to the number of GCN parameters. The experimental results are as follows Table 3-3.
>
> [Table 3-3] Test ACC on Cora for a fair comparison
> +---------------------------+----------------------+---------------------+-------------------+
> | Model/Test ACC      |          Cora          |       Citeseer      |     Pubmed    |
> +---------------------------+----------------------+---------------------+-------------------+
> | GCN[4]                      |  81.5% (23040) |  70.3% (56895) | 79.0% (8048) |
> | GSM (our base)       |  82.1% (21532) |  69.5% (57120) | 79.6% (8260) |
> +---------------------------+----------------------+---------------------+-------------------+
> (the number of parameters)
>
> GSM has reached or outperformed GCN even with a similar number of parameters as GCN. Using excessive parameters is a disadvantage of the model, as R2 pointed out. However, for GCN, using more parameters does not guarantee to improve performance. Although GSM uses more parameters, it adaptively reflects the local and global of graph information, which leads result to SOTA. Considering the results of Table 3-3 and Table 5 in the draft, we can conjecture that our model does not overfit the data but effectively uses the parameters for modeling graph structure to avoid the oversmoothing issue.

---

> ### Author Response · Authors · 2019-11-11
> **Author response to Review #2 (part3)**
>
> 4. As your work is quite similar to [1, 2], it would be beneficial to also include the respective results of those methods in Tables 2 and 3. In addition, their differences and similarities should be discussed in detail.
>
> A)
> The differences between our model and [1, 2, 3] are described in R2’s question #1, and as R2 mentioned, we added the experimental results of [1, 2, 3] on Table 2 and 3 . The updated results still confirm that our method outperforms the other methods. You will see this in the modified version.
>
>
> 5. Since the used benchmark datasets are already reasonably explored, authors are advised to include evaluation on other datasets as well, e.g., from [6].
>
> A)
> (Test ACC)
> +---------------------------------+------------------+--------------------------+-----------------------------+---------------------+
> | Model/Test ACC             | Coauthor CS | Coauthor Physics | Amazon Computers | Amazon Photo |
> +---------------------------------+------------------+--------------------------+-----------------------------+----------------------+
> | GCN[4]                             |     91.1±0.5   |           92.8±1.0         |            82.6±2.4           |        91.2±1.2      |
> | GAT[5]                              |     90.5±0.6   |           92.5±0.9         |            78.0±19.0         |        85.7±20.3    |
> | GSM (our base)              |     91.8±0.4   |           93.3±0.6         |            79.2±2.1           |        89.3±1.9       |
> | GESM (GSM+attention) |     92.0±0.5   |           93.7±0.6         |            79.3±1.7           |        90.0±2.0      |
> +---------------------------------+-----------------+---------------------------+-----------------------------+----------------------+
>
> We added the experiment R2 mentioned. We've been able to see that our model is pretty robust in a variety of datasets and environments.
>
>
> 6. The transition matrix is missing self-loops to match with the results of Figure 1. Since you already define analogously to , you should focus on one notation for consistency reasons.
>
> A)
> Thank the R2 for pointing out our mistake. We added self-loops to Figure 1.
>
>
> We will update our manuscript by reflecting the comments and responses as soon as possible.
>
> [1] Wu et al.: Simplifying Graph Convolutional Networks
> [2] Klicpera et al.: Predict then Propagate: Graph Neural Networks meet Personalized PageRank
> [3] Xu et al.: Representation Learning on Graphs with Jumping Knowledge Networks
> [4] Kipf and Welling: Semi-Supervised Classification with Graph Convolutional Networks
> [5] Petar Veličković et al: Graph Attention Networks
> [6] Shchur et al.: Pitfalls of Graph Neural Network Evaluation
> [7] Gleich et al. : Seeded PageRank solution paths
> [8]  Sami Abu-El-Haija et al. : MixHop: Higher-Order Graph Convolutional Architectures via Sparsified Neighborhood Mixing

---

### Official Review · AnonReviewer3 · 2019-10-31
**Official Blind Review #3**

**Rating:** 3

**Review:**

The paper proposes a new GNN model to address the common issue “oversmoothing”, namely, Graph Entities with Step Mixture via random walk (GESM). Basically, it integrates both mixture of various steps through random walk, and graph attention network, and demonstrates that it can overcome the SOTA on popular benchmarks.

Detailed comments:

* The oversmoothing problem has been mentioned many times in this paper, yet little has been demonstrated through experiments that the new model can solve the oversmoothing issue. It would be great to show the performance improvement while oversmoothing is mitigated.

* The proposed idea is very similar to the following paper: “Revisiting Graph Neural Networks: All We Have is Low-Pass Filters”. Both use low-pass filtering (via transition matrix) to propagate the information on the graph. I suggest a detailed discussion with this work.

* The major concern of this work is the weak novelty. It combines GAT with multiple random walk under the GNN framework. While this is working well on most GNN datasets, it is not very new by itself.

* Some experimental results are comparable to existing methods, as shown in Table 2 and 3. Maybe the time complexity is the major contribution of this paper. A head-to-head running time comparison with SOTA in Table 4 will be helpful.

* Fewer methods are compared in Table 3 than in Table 2. Can authors add more in Table 3 to give a better demonstration?


**Experience Assessment:**

I have published one or two papers in this area.

**Review Assessment: Checking Correctness Of Derivations And Theory:**

I assessed the sensibility of the derivations and theory.

**Review Assessment: Checking Correctness Of Experiments:**

I carefully checked the experiments.

**Review Assessment: Thoroughness In Paper Reading:**

I read the paper at least twice and used my best judgement in assessing the paper.

---

> ### Author Response · Authors · 2019-11-11
> **Author response to Review #3 (part1)**
>
> We appreciate R3 for the constructive feedback. Thanks to your comments, we found that there was a shortage of empirical proofs on oversmoothing. We agree with your opinion and thus conducted more experiments regarding oversmoothing.
>
> The responses for each comment are as follows:
>
>
> 1. The oversmoothing problem has been mentioned many times in this paper, yet little has been demonstrated through experiments that the new model can solve the oversmoothing issue. It would be great to show the performance improvement while oversmoothing is mitigated.
>
> A)
> About the oversmoothing issue, we have already presented experimentally that the accuracy does not degrade as the step increases in Figure 4b (Figure 5 in revision). However, as R3 pointed out, to give a more explicit demonstration, we compared ours with other competitive methods on how training accuracy changes depending on the step size. The results are as follows :
>
> (Train ACC)
> +-----------------+-------+-------+-------+-------+-------+
> | model/step |   2   |    5   |    8   |   15   |   20  |
> +-----------------+-------+-------+-------+-------+-------+
> | GCN[5]        | 1.00 | 1.00 | 0.92 | 0.23 | 0.20 |
> | GAT[7]         | 1.00 | 1.00 | 0.89 | 0.25 | 0.22 |
> | SGC[1]         | 1.00 | 1.00 | 0.87 | 0.77 | 0.72 |
> | GSM (ours) | 1.00 | 1.00 | 1.00 | 1.00 | 1.00 |
> +-----------------+-------+-------+-------+-------+-------+
>
> As shown in the above results, the other methods such as GCN[5], SGC[1], and GAT[7] suffer from oversmoothing; GCN and GAT show severe degradation in accuracy since the 8th step; SGC is better than GCN and GAT, but accuracy gradually decreases as the step size increases. These results means other models cannot train from the data due to the oversmoothing. Unlike others, the proposed GSM maintains performance without any degradation, because no rank loss[4] occurs and oversmoothing is overcome by step mixture.
>
>
>  2. The proposed idea is very similar to the following paper: “Revisiting Graph Neural Networks: All We Have is Low-Pass Filters”. Both use low-pass filtering (via transition matrix) to propagate the information on the graph. I suggest a detailed discussion with this work.
>
> A)
> Thank you for your insightful feedback. As R3 stated, there are similarities between our model and gfNN[8] in that propagation and embedding are separated. However, different from gfNN, we gather all the progressed blocks and consider all of them for the final prediction. This is what we call step-mixture, which allows our model to adaptively select global and local information from all graphs.
>
>
> 3. The major concern of this work is the weak novelty. It combines GAT with multiple random walk under the GNN framework. While this is working well on most GNN datasets, it is not very new by itself.
>
> A)
> (Inference Time(s))
> +-----------------------+---------+----------+---------+---------+---------+
> |    Model/Step     |     2     |     5     |     8      |   15    |   20    |
> +-----------------------+---------+----------+---------+---------+---------+
> | GCN[5]                | 0.028  | 0.037  | 0.049  | 0.073 | 0.092 |
> | GAT[7]                 | 0.136  | 0.201  | 0.315  | 0.577 | 0.781 |
> | GSM                    | 0.035  | 0.039  | 0.043  | 0.060 | 0.071 |
> | GESM                  | 0.131  | 0.143  | 0.153  | 0.178 | 0.211 |
> +-----------------------+---------+---------+----------+---------+---------+
>
> (Test ACC)
> +---------------------------------+------------------+--------------------------+---------------------+-----------------------+
> | Model/Test ACC             | Coauthor CS | Coauthor Physics | Amazon Computers | Amazon Photo |
> +---------------------------------+------------------+--------------------------+---------------------+-----------------------+
> | GCN[5]                             |     91.1±0.5   |           92.8±1.0         |    82.6±2.4         |        91.2±1.2       |
> | GAT[7]                              |     90.5±0.6   |           92.5±0.9         |    78.0±19.0       |        85.7±20.3    |
> | GSM (our base)              |     91.8±0.4   |           93.3±0.6         |    79.2±2.1         |        89.3±1.9       |
> | GESM (GSM+attention) |     92.0±0.5   |           93.7±0.6         |    79.3±1.7         |        90.0±2.0      |
> +---------------------------------+-----------------+---------------------------+----------------------+----------------------+
>
> We agree that the novelty of our method might not be very large. However, it is never trivial to integrate two methods to address the oversmoothing issue with smaller computation time. Our method provides much faster and more accurate performance compared to GAT[7], which can be done by the sophisticated design of our method using mixture of random walk steps. Our contribution is to robustly handle the oversmoothing issue while spending much smaller computation costs than GATs as shown in Table above (A new experiment is conducted on [6] by the suggestion of Reviewer 2).

---

> ### Author Response · Authors · 2019-11-11
> **Author response to Review #3 (part2)**
>
> 4. Some experimental results are comparable to existing methods, as shown in Table 2 and 3. Maybe the time complexity is the major contribution of this paper. A head-to-head running time comparison with SOTA in Table 4 will be helpful.
>
> A)
> Thank you for your valuable comments. We measured a head-to-head running time in terms of inference with GCN[5], GAT[7], and our GSM, GESM on Cora dataset as displayed in reply 3 and revised manuscript Figure 6. The running time of our base GSM model is comparable to very fast GCN, and our attention-enhanced GESM is much faster than GAT.
>
>
> 5. Fewer methods are compared in Table 3 than in Table 2. Can authors add more in Table 3 to give a better demonstration?
>
> A)
> Thank you for raising this issue. Only a few papers conducted experiments with low label rates, so a small number of methods were compared through public reports. For a better demonstration according to your comments, we added experimental results of SGC[1], APPNP[2], and JK-Net[3] by our own implementation. The updated Table 3 still confirms the competitiveness of our method.
>
>
> We will update our manuscript by reflecting the comments and responses as soon as possible.
>
> [1] Wu et al.: Simplifying Graph Convolutional Networks
> [2] Klicpera et al.: Predict then Propagate: Graph Neural Networks meet Personalized PageRank
> [3] Xu et al.: Representation Learning on Graphs with Jumping Knowledge Networks
> [4] Sitao Luan et al: Break the Ceiling: Stronger Multi-scale Deep Graph Convolutional Networks
> [5] Kipf and Welling: Semi-Supervised Classification with Graph Convolutional Networks
> [6] Shchur et al.: Pitfalls of Graph Neural Network Evaluation
> [7] Petar Veličković et al: Graph Attention Networks
> [8] Hoang NT and Takanori Maehara: Revisiting Graph Neural Networks: All We Have is Low-Pass Filters

---

### Author Response · Authors · 2019-11-12
**Revised draft uploaded**

Dear reviewers and all,

We have updated our draft according to the reviewers’ comments. We thank the reviewers for their rich insightful comments. We would not have been able to reach this revision without the reviewers’ advices. We believe our draft has improved significantly and would be very grateful if we are informed about any further concerns.

To summarize our main changes:

1. We have modified the last paragraph of page 3 in accordance with the advice of R1. This is also about JK-Net, which R2 pointed out.

2. We ran all the experiments and added the results in Section 5.2. The contents of Section 5.2 now include an explanation of oversmoothing (which R3 pointed out), the differences between JK-Net, SGC and our models (which R2 pointed out), and the inference time that all the reviewers pointed out. Overall, we have been able to verify that our model outperforms existing models in many aspects.

3. We added the results of JK-GCN, APPNP, SGC in Table 2 and 3  in accordance with the advice of R3. Our models (GSM and GESM) still outperforms existing models.

4. We inserted experiments on additional datasets (which R2 proposed) and visualization of attention distributions (which R1 proposed) to the Appendix. We have been able to confirm that our model does not overfit on a particular dataset and move a step closer to understanding the effect of attention through visualizing the attention distribution.

Kind regards,
Authors

---

### Decision · Program_Chairs · 2019-12-19

**Decision:**

Reject

**Comment:**

Two reviewers are concerned about this paper while the other one is slightly positive. A reject is recommended.